# Impaired Morphogenesis and Function of Rat Adrenal Zona Glomerulosa by Developmental Low-Dose Exposure to DDT Is Associated with Altered Oct4 Expression

**DOI:** 10.3390/ijms22126324

**Published:** 2021-06-12

**Authors:** Nataliya V. Yaglova, Sergey S. Obernikhin, Dibakhan A. Tsomartova, Svetlana V. Nazimova, Valentin V. Yaglov, Elina S. Tsomartova, Elizaveta V. Chereshneva, Marina Y. Ivanova, Tatiana A. Lomanovskaya

**Affiliations:** 1Laboratory of Endocrine System Development, Federal State Budgetary Scientific Institution “Research Institute of Human Morphology”, 117418 Moscow, Russia; ober@mail.ru (S.S.O.); dtsomartova@mail.ru (D.A.T.); pimka60@list.ru (S.V.N.); vyaglov@mail.ru (V.V.Y.); tselso@yandex.ru (E.S.T.); 2Department of Histology, Cytology, and Embryology, Federal State Funded Educational Institution of Higher Education I.M. Sechenov First Moscow State Medical University, 119435 Moscow, Russia; yelizaveta.new@mail.ru (E.V.C.); ivanova_m_y@mail.ru (M.Y.I.); tatiana_80_80@inbox.ru (T.A.L.)

**Keywords:** Oct4, morphogenesis, endocrine disrupting chemicals, DDT, adrenal gland, zona glomerulosa

## Abstract

Dichlorodiphenyltrichloroethane (DDT) is a persistent organic pollutant and one of the most widespread endocrine disrupting chemicals. The impact of low-dose exposure to DDT on the morphogenesis of the adrenal gland is still poorly understood. The development and function of zona glomerulosa in rats has been found to be associated with changes in the expression of the transcription factor Oct4 (Octamer 4), which is the most important player in cell pluripotency. The aim of the study was to investigate the morphogenesis and function of rat adrenal zona glomerulosa in rats exposed to low doses of DDT during prenatal and postnatal development and to determine the possible role of Oct4 in DDT-mediated structural and functional changes. The DDT-exposed rats demonstrated slower development and lower functional activity of the zona glomerulosa during the pubertal period associated with higher expression of Oct4. Further, accelerated growth and restoration of hormone production was associated with, firstly, a decrease in Oct4 expressing cells and secondly, the loss of the inverse relationship between basal aldosterone levels and the number of Oct4 expressing cells. Thus, the transcriptional factor Oct4 exhibited an altered pattern of expression in the DDT-exposed rats during postnatal development. The results of the study uncover a novel putative mechanism by which low doses of DDT disrupt the development of adrenal zona glomerulosa.

## 1. Introduction

Incidences of endocrine-related disorders and associated somatic diseases have been increasing in recent decades. There is a growing body of evidence for negative effects of endocrine disrupting chemicals on human health and for the contribution of endocrine disruption to a rise in endocrine-related pathology [1,2]. According to the World Health Organization, endocrine disrupting chemicals are substances that alter one or more functions of the endocrine system and consequently cause adverse health effects in an intact organism, or its progeny, or (sub)populations [1]. The Endocrine Society was the first to draw attention to the endocrine disrupters, considering them a novel direction for scientific investigations [2]. Since the first Scientific Statement of the Endocrine Society in 2009, data on animal and human studies have demonstrated the harmful effects of endocrine disrupters on the function and development of an organism [3,4,5,6]. Large-scale investigations on the impact of endocrine disrupters on morphogenetic processes, prenatal and postnatal histogenesis and organogenesis are required to decipher the mechanisms of congenital and developmental disorders and to revisit our knowledge on endocrine regulation of metabolism.

There are hundreds of chemical compounds affecting endocrine function [7,8]. The intensity of their effects on the body is determined primarily by the route, dose and duration of exposure. One of the main sources of nonoccupational exposure to endocrine disrupting chemicals is food products.

Dichlorodiphenyltrichloroethane (DDT) is an organochlorine pesticide. Extensive use of DDT in the 20th century and the resumption of use in the 21st century for disease vector control, and long half-life have made it a universal persistent pollutant and food contaminant [9]. Maximal permissive levels for DDT in food range from 50 up to 400 μg/kg and a provisional tolerable daily intake of DDT and its associated compounds is 0.01 mg/kg bw [10]. DDT is a highly lipophylic substance with a low metabolic rate [11]. Because of its lipophylicity, it accumulates in the different tissues of the human body [12]. The low excretion rate of DDT and its metabolites lead to a prolonged disruption of endocrine function. Investigations have showed that DDT acts as an androgen receptor antagonist [13] and demonstrates an estrogen-mimicking effect [14]. Epidemiological investigations have revealed higher incidences of precocious puberty, malformations and impaired functions of the reproductive systems in both sexes in regions where DDT was used as a pesticide in agriculture [15,16,17,18]. Recent studies have showed the negative effect of nonoccupational exposure to DDT on thyroid function [19,20]. The impact of low-dose exposure to DDT on morphogenesis of the adrenal gland is less studied. 

Numerous studies demonstrate that endocrine disruptors can alter the epigenetic regulation of morphogenetic processes [21,22,23]. In our previous studies, we found downregulation of canonical Wnt-signaling in the adrenal cortical cells, which was most pronounced in the zona glomerulosa during postnatal development in rats exposed to low doses of DDT [24]. Alteration in the development and function of the adrenal cortex suggests an impaired proliferation and differentiation of the adrenal cortical cells with the implication of the transcription factors regulating morphogenesis.

Some investigations have revealed that stem and pluripotent cells are retained in postnatal endocrine glands and provide a source for cell turnover and reparation [25]. The transcriptional factor Oct4 (Octamer-binding transcription factor 4, also known as Otf3 or Oct3/4) plays a key role in maintaining pluripotency and in the differentiation of embryonic cells [26]. Oct4, a homeodomain transcription factor of the POU family, together with Sox2, Klf4, c-Myc, also known as Yamanaka factors, are highly expressed in embryonic stem cells. Their overexpression has been found to induce pluripotency in both mouse and human somatic cells [27]. The role of Oct4 in postnatal histogenesis is obscure. Single reports demonstrate Oct4 expression in different types of cells in an adult organism [28,29]. In our previous studies we have found Oct4 expression in the adrenal cortex of a rat [30,31]. The zona glomerulosa demonstrated the highest rate of Oct4 expression and there was a strong inverse correlation between the number of Oct4-producing cells and the production of aldosterone. These findings suggest that impaired Oct4 signaling may be a putative mechanism of the dismorphogenetic effects of endocrine disrupting chemicals. The present study aimed to evaluate postnatal morphogenesis and the function of the adrenal zona glomerulosa in rats exposed to low doses of DDT during prenatal and postnatal development and aimed to elucidate the implication of the transcription factor Oct4 in the endocrine disruption of the development of the zona glomerulosa and hormone production.

## 2. Results

### 2.1. Histology of the Zona Glomerulosa of the Adrenal Cortex

The zona glomerulosa in the control pubertal rats was clearly recognized as a regular concentric layer beneath the capsule. It was composed of ovoid shaped cells. The nuclei were round or oval. Cytoplasm was enlightened due to numerous lipid droplets. The cells were arranged in oval clusters separated by thin trabeculae and capillaries. Capillaries in the zona glomerulosa were filled with plasma and did not contain blood cells (Figure 1A). 

Pubertal rats exposed to low doses of DDT demonstrated defragmentation and a narrowing of the zona glomerulosa. A histological examination also revealed typical signs of impaired microcirculation, such as the dilatation of microcirculatory vessels and obturation with red blood cells (Figure 1B). The surface area of the zona glomerulosa in equatorial sections was diminished compared to the control (Figure 1E). The glomerulosa cells were smaller (Figure 1F) and some cells exhibited picnotic nuclei. 

**Figure 1 ijms-22-06324-f001:**
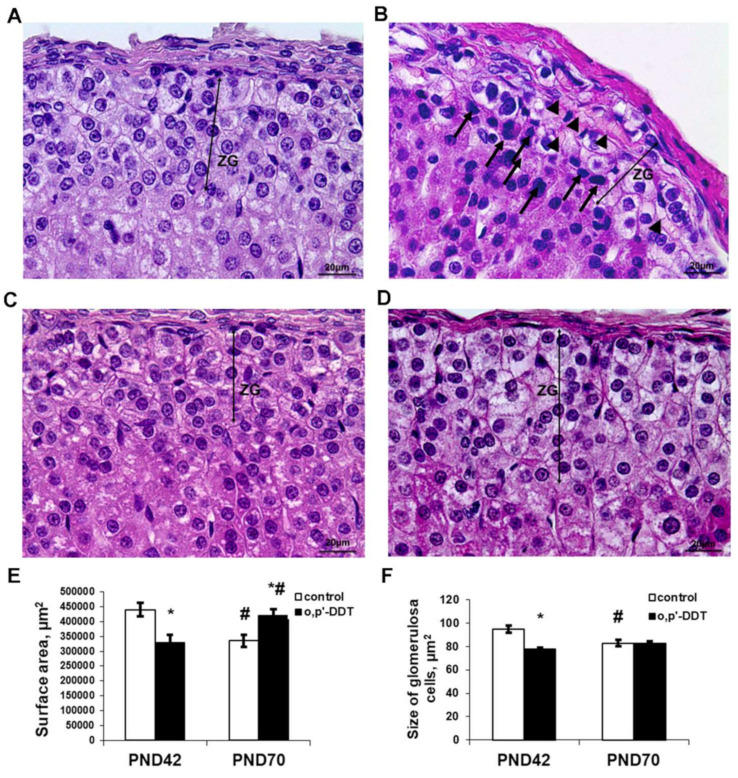
Changes in the histology of the adrenal zona glomerulosa after developmental exposure to low doses of o,p’-DDT. Histology of the zona glomerulosa (ZG) of the control rats (**A**) and DDT-exposed rats (**B**) in pubertal period (PND42), of the control rats (**C**) and DDT-exposed rats (**D**) after puberty (PND70). Arrowheads point to microcirculatory vessels obturated red blood cells with, arrows–picnotic nuclei. Scale bar 20 µm. (**E**) Surface area of the zona glomerulosa. (**F**) Size of the glomerulosa cell. Data are shown as mean ± S.E.M. PND, day of postnatal development; *p* < 0.05 compared to the control (*), compared to PND42 (#).

After puberty, the control rats showed a reduction in the zona glomerulosa with no principal changes in structure (Figure 1C,E,F). Unlike the control, DDT-exposed rats demonstrated an enlargement of the zona glomerulosa after puberty, and the surface area of their zona glomerulosa exceeded the control values (Figure 1D,E). The major mechanism of its enlargement was the restoration of the integrity of the concentric structure absent in the previous term. No microcirculatory disorders were observed.

### 2.2. Proliferative Activity of Glomerulosa Cells

An immunohistochemical evaluation of Ki-67 expression revealed high proliferative activity of the glomerulosa cells in the pubertal period and its two-fold decrease after puberty on the 70th day of postnatal development in the control rats (Figure 2). The proliferation rate of the glomerulosa cells in pubertal DDT-exposed rats was significantly higher than in the control rats. After puberty, no differences in the expression of Ki-67 between the exposed and control rats were found (Figure 2).

### 2.3. Aldosterone Production

The serum level of aldosterone in DDT-exposed rats in the pubertal period was significantly lower than in the control rats (Figure 3). After puberty, both the control and DDT-exposed rats had a higher level of aldosterone production than in puberty, and no differences in aldosterone serum levels were observed (Figure 3).

### 2.4. Expression of Transcriptional Factor Oct4

Immunohistochemical detection revealed the nuclear localization of Oct4 in glomerulosa cells (Figure 4A). In the pubertal period, Oct4 exhibited a focal pattern of expression in the zona glomerulosa of the control rats: single cells with Oct4-positive nuclei were observed in the neighboring glomeruli while some areas of the zona glomerulosa were Oct4-negative. The number of Oct4-positive cells in the zona glomerulosa of DDT-exposed rats significantly exceeded the control values in the pubertal period (Figure 4B,E). Oct4 demonstrated a diffuse pattern of expression.

After puberty, the number of the Oct4-positive cells in the zona glomerulosa of the control rats was two times higher than in the pubertal period (Figure 4C,E). DDT-exposed rats demonstrated a down-regulation of Oct4 expression with age and a lower level of Oct4-positive cells compared to the control (Figure 4D,E).

A strong correlation between the number of Oct4-positive glomerulosa cells and aldosterone serum content was found in the pubertal control (R = −0.98 *p* = 0.000001) and DDT-exposed rats (R = −0.95 *p* = 0.000002). After puberty the interdependence between these parameters in the control rats decreased (R = −0.83 *p* = 0.0058) and was statistically insignificant in the DDT-exposed animals (R = 0.3, *p* = 0.65).

**Figure 4 ijms-22-06324-f004:**
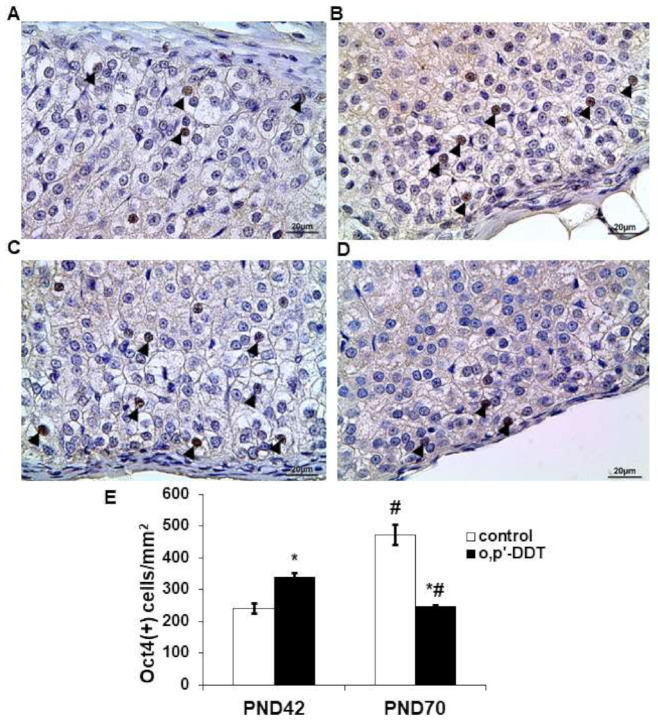
Changes in expression of Oct4 in glomerulosa cells after developmental exposure to low doses of o,p’-DDT. Immunohistochemical detection in the zona glomerulosa of the control (**A**) and DDT-exposed (**B**) rats in pubertal period (PND42) and of the control (**C**) and DDT-exposed (**D**) rats after puberty (PND70). Arrowheads point to Oct4-positive cells in the zona glomerulosa. Scale bar 20 µm. (**E**) Number of Oct4-positive cells in 1 mm^2^ of the zona glomerulosa; Data are shown as mean ± S.E.M. PND, day of postnatal development; *p* < 0.05 compared to the control (*), compared to PND42 (#).

## 3. Discussion

The present investigation demonstrates that developmental exposure to low doses of DDT alters the postnatal development of the zona glomerulosa. Normal postnatal morphogenesis of the zona glomerulosa, unlike zona fasciculata, exhibit dual phases during the growth of the adrenal cortex. Active growth before sexual maturation is followed by shrinkage after puberty [32]. In our investigation the control rats also demonstrated a higher surface volume of the zona glomerulosa in pubertal age than after puberty. Termination of growth was associated with a decrease in the proliferation rate and an increase in the number of Oct4-positive cells. Despite the fact that the role of Oct4 in the self-renewal of somatic cells is controversial [33], Oct4-expressing cells in the adult adrenal cortex are most likely required for cell renewal since the reparation of the lesions in the zona fasciculata has been found to be associated with an up-regulation of Oct4 expression in the adrenal cortex [34]. Expression of Oct4 in glomerulosa cells negatively correlated with the aldosterone serum level, suggesting that Oct4 is implicated in the regulation of aldosterone production both in the pubertal period and after puberty.

A histological examination of the zona glomerulosa and an evaluation of cell proliferation activity in DDT-exposed animals revealed changes that are indicative of a lower rate of postnatal development. We also observed an obturation of the microcirculatory vessels and the subsequent signs of hypoxia, such as smaller sizes of the cells with picnotic nuclei. Unlike control animals, DDT-exposed rats demonstrated an accelerated growth of the zona glomerulosa between PND42 and PND70. The pattern of Oct4 expression also changed. A higher number of Oct4-positive cells in the pubertal period was associated with a smaller size of the zona glomerulosa and its focal growth, which is indicative of an earlier stage of development, and insufficient aldosterone production. Altered Oct4 expression in pubertal rats could result from two factors, aggravating each other: the direct action of the endocrine disruptor and an impaired oxygen supply due to microcirculatory disorders. Recent studies have shown that cellular hypoxia induces the expression of Oct4 [35,36], and therefore, these two factors could mutually enhance each other’s interference in the development of the zona glomerulosa. We suppose that higher expression of Oct4 in DDT-exposed pubertal rats reflects, to a greater extent, the delayed development of the zona glomerulosa, rather than cell hypoxia induced by impaired blood circulation. Hypoxia is also known to inhibit aldosterone synthesis [37,38] and may be considered as an additional factor implicated in decreased aldosterone production. However, the negative correlation between the aldosterone serum concentration and the number of Oct4-expressing cells suggests the implication of Oct4 in delayed growth and the functional maturation of the zona glomerulosa. The zona glomerulosa is known to be a source for cell renewal throughout the adrenal cortex, due to the presence of low-differentiated cells and cells capable of transdifferentiation from glomerulosa to fasciculata phenotype [39,40]. The active growth of the zona glomerulosa after puberty led to a depletion of the Oct4-positive cell pool in adult DDT-exposed rats. It also indicates that Oct4 is implicated in growth and differentiation control, and DDT disrupts postnatal growth by interfering in the Oct4-mediated maintenance of the pluripotent cells.

It is noteworthy that the relative hyperplasia of the zona glomerulosa in adult DDT-exposed rats did not result in the hyperproduction of aldosterone. The absence of a correlation between Oct4 expression and aldosterone serum content indicates that the disruption of aldosterone production is not only attributed to diminished Oct4 expression and suggests an additional mechanism of relative functional insufficiency besides the Oct4-provided maintenance of pluripotent potential. The possible disruption of thyroid function may also contribute to impaired postnatal development by interfering in transcriptional regulation of cell functional maturation, since thyroid hormones are well-known promoters of cell differentiation [40,41]. Numerous studies have reported effects on thyroid function and neurodevelopment by prenatal DDT exposure, suggesting that the inhibition of thyroid hormone production could be a putative mechanism underlying morphogenetic disorders [42,43,44,45]. Our results show that the restoration of zona glomerulosa integrity and growth seem to be the main reasons for higher aldosterone synthesis in DDT-exposed rats in puberty. The mechanisms of compensatory growth activation are not clear since the exposure of rats to DDT continued. The accelerated growth of the zona glomerulosa due to the activation of cell proliferation and upregulation of Oct4 expression found in the present study supports the hypothesis that the transcription factor Oct4 is implicated in the control of postnatal development and may be considered a target of endocrine disrupters.

## 4. Materials and Methods

### 4.1. Animals

Female and male Wistar rats were obtained from the Scientific Center of Biomedical Technologies of Federal Medical and Biological Agency of Russia. The rats were housed at +22–23 °C and given a pelleted standard chow ad libitum. The investigation was performed in accordance with the handling standards and rules of laboratory animals as consistent with “International Guidelines for Biomedical Researches with Animals” (1985), laboratory routine standards in the Russian Federation (Order of Ministry of Healthcare of the Russian Federation dated 19 June 2003 No. 267) and “Animal Cruelty Protection Act” dated 1 December 1999, and the regulations of experimental animal operation approved by Order of Ministry of Healthcare of USSR No. 577 dated 12 August 1977. The animal procedures were approved by the Ethics Committee of the Research Institute of Human Morphology (protocol N8a, 03.09.2015).

### 4.2. Experimental Design

The female rats that weighed 180–220 g received a solution of o,p-DDT 20 µg/L (“Sigma-Aldrich”, St. Louis, MO, USA) ad libitum instead of tap water since mating during pregnancy and lactation. After weaning the progeny of the rat dams received the same solution of o,p-DDT during postnatal development. The progeny of intact female rats were used as a control. Only male rats were used for examination (20 DDT-exposed and 20 control rats). The rats were sacrificed by zoletil overdosage in the pubertal period on the 42nd day of postnatal development (PND42) and after puberty on the 70th day of their postnatal development (PND70) when rat adrenals reach their maximal development [46]. The average daily intake of DDT after weaning was 2.90 ± 0.12 µg/kg bw, which corresponded to the DDT consumption by humans with food products with consideration for the differences in DDT metabolism in rats and humans [47]. The absence of DDT, its metabolites, and related organochlorine compounds in tap water and chaw was confirmed by gas chromatography in the Moscow Federal Budgetary Institution of Public Health. Blood and adrenal glands were collected.

### 4.3. Adrenal Histology

The adrenal glands were fixed in Bouen solution. After standard histological processing, the tissue samples were embedded in paraffin. Equatorial sections of the adrenals were stained with hematoxylin and eosin. A histological examination was performed with a “Leica DM2500” light microscope (Leica Microsystems Gmbh, Wetzlar, Germany). The histological examination included light microscopy and computer morphometry of the zona glomerulosa.

Computer morphometry of the light microscope images was carried out using “ImageScope” software (Leica Microsystems Gmbh, Wetzlar, Germany). The surface area of the zona glomerulosa and the size of the glomerulosa cells were measured.

### 4.4. Immunohistochemistry

An immunohistochemical evaluation of Ki-67 and Oct4 was performed on the paraffin-embedded tissues. After antigen retrieval with 10 mM sodium citrate (pH 6.0), endogenous peroxidase and endogenous immunoglobulins were blocked with Hydrogen Peroxide Block and Protein Block (Thermo Fisher Scientific, Waltham, MA, USA). The slides were incubated with primary antibodies to Ki-67 (1:100, Cell Marque, Rocklin, CA, USA) and Oct4 (1:5000, Abcam, Cambridge, MA, USA) overnight at 8 °C. Sections of rat embryonic tissues were used as a positive control for Oct4. The slides processed without incubation with primary antibodies were used as a negative control. The reaction was visualized with an UltraVision LP Detection System reagent kit (Thermo Fisher Scientific, Waltham, MA, USA) according to the manufacturer’s recommendations. The sections were counterstained with Mayer’s hematoxylin.

Expression of Oct4 and Ki-67 was assessed as number of immunopositive cells with nuclear staining per 1 mm^2^ both on the 42nd and 70th days because no difference in cellular density of the adrenal cortices in pubertal and postpubertal rats was found.

### 4.5. Enzyme-Linked Immunosorbent Assay

Aldosterone serum levels were determined by ELISA with a rat aldosterone kit (Cusabio, Wuhan, China). The optic density was measured with an “Anthos 2010” microplate reader at 450 nm.

### 4.6. Statistical Analysis

The statistical analyses were carried out using the software package Statistica 7.0 (StatSoft, Palo Alto, CA, USA). The central tendency and dispersion of the quantitative traits with approximately normal distribution were presented as the mean and standard error of the mean (M ± SEM). The association of the number of Oct4-positive glomerulosa cells and the aldosterone serum concentration were performed with Pearson correlations. Quantitative comparisons of the independent groups were performed using Student’s t-test taking into account the values of Levene’s test for the equality of variances. The differences were considered to be statistically significant at *p* < 0.05.

## 5. Conclusions

In summary, the present study revealed that prenatal and postnatal exposure to low doses of DDT affects cell proliferation and differentiation and attenuates the development and function of the zona glomerulosa. The lower degree of development and aldosterone production found in the pubertal period correlated with overexpression of Oct4. The active compensatory growth of the zona glomerulosa and the restoration of the hormone production rate after puberty were associated with a reduction in Oct4-expressing cells. Our results show that Oct4 may be considered a target of endocrine disrupters and they have uncovered an implication of the transcription factor Oct4 in mechanisms by which low doses of DDT disrupt the development of the adrenal zona glomerulosa.

## Figures and Tables

**Figure 2 ijms-22-06324-f002:**
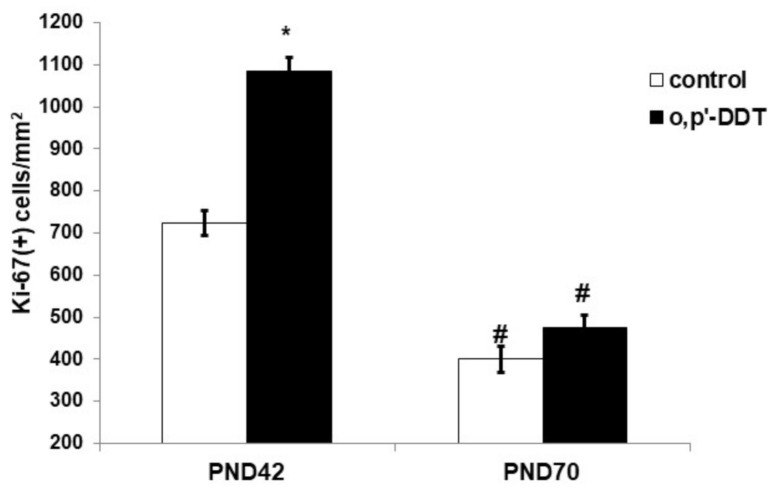
Proliferative activity of the glomerulosa cells in the control and o,p-DDT exposed rats. Data are shown as mean ± S.E.M. PND, day of postnatal development; *p* < 0.05 compared to the control (*), compared to PND42 (#).

**Figure 3 ijms-22-06324-f003:**
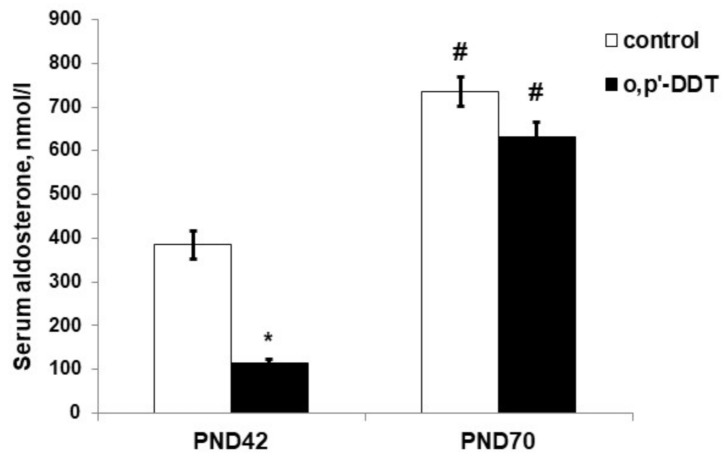
Aldosterone serum concentration in the control and o,p-DDT exposed rats. Data are shown as mean ± S.E.M. PND, day of postnatal development; *p* < 0.05 compared to the control (*), compared to PND42 (#).

## Data Availability

The data presented in this study are available from the corresponding author upon reasonable request.

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
