# Peer review of "Impaired Morphogenesis and Function of Rat Adrenal Zona Glomerulosa by Developmental Low-Dose Exposure to DDT Is Associated with Altered Oct4 Expression"

_ijms, 2021, doi:10.3390/ijms22126324_

Round 1
Reviewer 1 Report
Yagloval et al described change in adrenal zona glomerulosa under DDT treated rats. DDT changed morphology of the zona glomerulosa (ZG), Oct4 expression, and aldosterone production. DDT (or DDD) is well known as adrenal toxin so it was very interesting to see the change in rat adrenal with their exposure. The paper was written well on what the authors did so that the paper is easy to read for readers. However, what the authors are showing only morphology and Oct4 expression. Probably, only these observation (ie: morphology and immuno) are not sufficient to come to their conclusions. The reviewer is wondering why the authors more detailed experiment including RNAseq or proteomics.. The experimental animals are solidly established as the morphology is apparently interesting. The impact of the current paper is too weak for pub in IJMS.
Author Response
Response 1: We are very grateful for all your comments.
In the present investigation we performed not only morphology and Oct4 expression. Proliferation and aldosterone production were also studied. We aimed at assessment of Oct4 expression in glomerulosa cells and chose immunohistochemistry because it is the only method that allows to determine type of Oct4-producing cells and their location (glomerulosa cells, subcapsular progenitors, or low-differentiated cells of zona intermedia) as well as intracellular localization of the protein. Due to immunohistochemical evaluation we determined that Oct4-positive cells are typical glomerulosa cells integrated into glomeruli, not progenitor cells, and Oct4 translocates into their nuclei. Oct4 expression in glomerulosa cells and its association with their functional activity, found in the present and previous studies, open up new directions for research on the role of Oct4 in steroidogenesis, cell renewal of zona glomerulosa, transdiiferentiation of glomerulosa cells into other adrenocortical cell types and possible role in tumorigenesis. Therefore, we hope that our report will be of interest to researchers in the fields of histology, pathology, endocrinology.
Reviewer 2 Report
PDF format
Impaired morphogenesis and function of rat adrenal zona glomerulosa by developmental low-dose exposure to DDT is associated with altered Oct4 expression.
Line 21: in pubertal period or should say during the puberal period.
Line 22-24: was a little confusing for me. Suggestion: Further, accelerated growth and restoration of hormone production was associated with; first, a decrease in Oct4 expressing cells and second, loss of the inverse relationship between basal aldosterone levels and the number of Oct4 expressing cells.
Line 32: However, you are investigating toxicology not necessarily somatic disease.
Line 33: After a statement like… “there is a growing body of evidence … you may need to add a reference or two.
Line 47: As well as dose, duration of exposure and exposure during prenatal critical period (Wilson’s 6 principles of teratogenesis)
Line 55: Unsure what you mean by low metabolic rate? Are you saying anything about the clearance from the body? Since DDT is lipophilic, the substance accumulates in the adipose tissue and thus is shielded from the liver and kidney.
Line 56: You are alluding to how low renal clearance rate contribute to the half life of the toxin and thus the duration of exposure.
Line 57: Superb thought here! DDT acting as an endocrine disrupter as an androgen antagonist (zona reticularis especially in females) and as an estrogenic substance.
Overall here you may state that DDT cause endocrinological disruption is not new in that the reproductive function (both male and females) as well as the thyroid function are profoundly affected by DDT. I can not help but notice the similarities in these organs either follicular (ovarian follicle, thyroid follicles) cord like configuration (seminiferous tubule (testis) and the zona granulosa (adrenal). What does DDT do the renal tubules?
Line 64: I would be tempted to put in a paragraph break here as you are now talking about the “genetics”.
Line 70: I suspect the zona fasciculata and reticularis are also affected based on reproductive disruption, especially in females.
Line 71: Are these cells primitive mesenchyme or from neural crest?
Line 88: Are you saying that Oct4 maintains the aldosterone secreting cells in a pluripotential state, and this poses a barrier for the secretion of aldosterone that would normally come from a differentiated cell.
Line 101: Since many disciplines will read this, it would be helpful to point out picnotic nuclei
Line 117: Is Ki-67 a marker for proliferating cells?
Line 127: Were these baseline Aldosterone levels? What would happen with stimulated levels such as hyperkalemia? Would the Oct4 promoter be involved?
Line 133: Figure 3
Line 138: Figure 4A
Line 142: Figure 4B and E
Figure 4 legend: the positive cells of the immunohistochemistry are indicated by the darker staining nuclei for Oct4.
Line 164: Do the cells themselves shrink as annotated in Figure 1F or is there also Apoptosis going on?
Line 177 with picnotic nuclei versus and picnotic nuclei ; Unlike control animals,
Line 184-185: Recent studies have shown that cellular hypoxia induces the expression of Oct4.
Line 205: Could one use another marker for glomerular cell maturation. It is possible that an increase in proliferation disunited with hormone secretion maybe due to immature cells increasing, not capable of aldosterone production. What would the organelles look like? Would there be aldosterone precursors in the cytoplasm?
Overall, I find this paper intriguing, you may want to put some time into presenting the argument of endocrine disruption as this important in the world of toxicology. I thank you for your efforts and take my comments as suggestions. I have been a cellular endocrinologist, worked in the area of embryology and also in the area of toxicology.
Author Response
Response to Reviewer 2
We are very grateful for all your comments and attention to the manuscript!
We agree with your comments regarding lines 21, 22-24, 33, 47, 55, 56, 57, 64, 133, 138, 142, 164, 177, 184. We have made corrections to the text according to your comments.
Reviewer comment 1: Overall here you may state that DDT cause endocrinological disruption is not new in that the reproductive function (both male and females) as well as the thyroid function are profoundly affected by DDT. I can not help but notice the similarities in these organs either follicular (ovarian follicle, thyroid follicles) cord like configuration (seminiferous tubule (testis) and the zona granulosa (adrenal). What does DDT do the renal tubules?
Response 1: Affection of renal tubules by DDT is obscure. No renal lesions have been reported even in case of severe DDT intoxication.
Reviewer comment 2: Line 70: I suspect the zona fasciculata and reticularis are also affected based on reproductive disruption, especially in females.
Response 2: According to our data, DDT significantly changes development and function of zona reticularis [Yaglova, N.V., Obernikhin, S.S., Yaglov, V.V., Nazimova S.V., Timokhina, E.P., Tsomartova, D.A. Low-Dose Exposure to Endocrine Disruptor Dichlorodiphenyltrichloroethane (DDT) Affects Transcriptional Regulation of Adrenal Zona Reticularis in Male Rats. Bulletin of Experimental Biology and Medicine, 2021, 170(5), стр. 682–685. DOI: 10.1007/s10517-021-05132-4]. Zona fasciculata development is less affected.
Reviewer comment 3: Line 71: Are these cells primitive mesenchyme or from neural crest?
Response 3: progenitor cells are considered of mesenchymal origin.
Reviewer comment 4: Line 88: Are you saying that Oct4 maintains the aldosterone secreting cells in a pluripotential state, and this poses a barrier for the secretion of aldosterone that would normally come from a differentiated cell.
Response 4: I think Oct4 drives glomerulosa cell to dedifferentiation and possibly to further transdifferentiation to other cortical cell types. It seems to be an additional source of growth and cell renewal for both zona glomerulosa and zona fasciculata.
Reviewer comment 5: Line 101: Since many disciplines will read this, it would be helpful to point out picnotic nuclei
Response 5: We marked picnotic nuclei with arrors.
Reviewer comment 6: Line 117: Is Ki-67 a marker for proliferating cells?
Response 6: Yes. It marks any stage of mitosis.
Reviewer comment 7: Line 127: Were these baseline aldosterone levels? What would happen with stimulated levels such as hyperkalemia? Would the Oct4 promoter be involved?
Response 7: Yes, we measured baseline aldosterone secretion.
Reviewer comment 8: Figure 4 legend: the positive cells of the immunohistochemistry are indicated by the darker staining nuclei for Oct4.
Response 8: Immunopositive cells have brown nuclei since DAB was used as a chromogen.
Reviewer comment 9: Line 164: Do the cells themselves shrink as annotated in Figure 1F or is there also Apoptosis going on?
Response 9: The cells decrease in size with age and reorganize mitochondrial apparatus.
Reviewer comment 10: Line 205: Could one use another marker for glomerular cell maturation. It is possible that an increase in proliferation disunited with hormone secretion maybe due to immature cells increasing, not capable of aldosterone production. What would the organelles look like? Would there be aldosterone precursors in the cytoplasm?
Response 10: Some amounts of aldosterone precursors like pregnanolone are always found in cytoplasm of glomerulosa cells. In our previous studies we found that functional maturation of glomerulosa cells during postnatal period of ontogeny is mainly due to reorganization of mitochondrial apparatus. DDT was found to prevent restructuration of mitochondria [N.V. Yaglova, S.S. Obernikhin, V.V. Yaglov, E.P. Timokhina, S.V. Nazimova, D.A. Tsomartova. Age-dependent changes of mitochondrial structure regulate steroidogenic activity of rat adrenal cortical cells. Clinical and Experimental morphology. 2020;9(1):64–70. DOI:10.31088/CEM2020.9.1.64-70].
Round 2
Reviewer 1 Report
The paper has been well updated.